# FastLane: Efficient Routed Systems for Late-Interaction Retrieval

## Abstract

Late-interaction retrieval models like ColBERT achieve superior accuracy by enabling token-level interactions, but their computational cost hinders scalability and integration with Approximate Nearest Neighbor Search (ANNS). We introduce FastLane, a novel retrieval framework that dynamically routes queries to their most informative representations, eliminating redundant token comparisons. FastLane employs a learnable routing mechanism optimized alongside the embedding model, leveraging self-attention and differentiable selection to maximize efficiency. Our approach reduces computational complexity by up to 30x while maintaining competitive retrieval performance. By bridging late-interaction models with ANNS, FastLane enables scalable, low-latency retrieval, making it feasible for large-scale applications such as search engines, recommendation systems, and question-answering platforms. This work opens pathways for multi-lingual, multi-modal, and long-context retrieval, pushing the frontier of efficient and adaptive information retrieval.

## 1 Introduction

How can we design systems that accurately and efficiently retrieve relevant information while capturing the nuanced meanings of queries? This question lies at the heart of modern information retrieval (IR), a field critical to applications such as search engines, question-answering systems, and recommendation platforms (Johnson et al., 2019; Nayak, 2019; Dahiya et al., 2021); retrieval augmented generation (Lewis et al., 2020); and so much more. Traditional IR methods often rely on single-vector representations of queries and documents. While these approaches are computationally efficient, they frequently struggle to capture the multi-faceted and nuanced semantic relationships present in natural language (Menon et al., 2022; Neelakantan et al., 2022; Muennighoff, 2022). For instance, consider the query, "what is the cost of apple?"; it remains ambiguous whether the question refers to the fruit or one of the various products such as phones and laptops. This example highlights the limitations of single-vector approaches in disambiguating and contextualizing semantic meaning, particularly when queries can have multiple interpretations. This problem arises in many cases of "branding", where multiple meanings for the same word, and the true intent of the user would be harder to gauge (see Figure 1).

Prior approaches (Khattab & Zaharia, 2020) have worked towards mitigating these limitations with the use of richer representations. Late-interaction methods represent each query token with an embedding and compute similarity using token-level Max-Sim interactions (depicted in Figure 3), rather than collapsing the sequence into a single pooled embedding (e.g., ["CLS"]) which is used for one dot-product similarity. Late-interaction methods offer a powerful solution to this challenge by representing a "searched query" as multiple embedding vectors. State-of-the-Art (SOTA) late-interaction models such as ColBERT (Khattab & Zaharia, 2020) leverage late interaction (sum-max, or other scoring mechanisms) to derive the relevance score between the query, and the document, leading to consistently improved performance over single-view dense retrieval methods (Menon et al., 2022; Neelakantan et al., 2022; Muennighoff, 2022).

Late-interaction models require additional computation, and thus are slower, which limits their widespread adoption. This cost in latency arises since late-interaction models require pairwise comparisons between all "query" and "target" views, instead of the condensed ["CLS"] view alone. This computation limits the

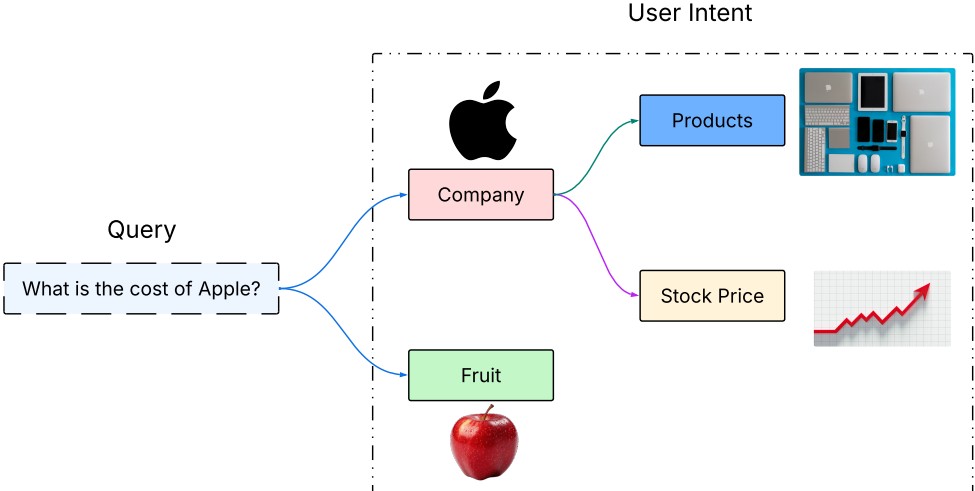

Figure 1: The query "What is the cost of Apple?" demonstrates how a single phrase can map to diverse user intents: i.e, from stock prices to fruit depending on context. This underscores the need for fine-grained token-level understanding to disambiguate meaning, especially as conventional NLP preprocessing (e.g., stemming, lemmatization) may obscure critical distinctions by dropping stopwords such as "a", "an", etc.

scalability, especially for large databases, as the system is no longer compatible with efficient Approximate Nearest Neighbor Search (ANNS) methods (Johnson et al., 2019; Sivic & Zisserman, 2003; Guo et al., 2020). Thus, most systems in production continue to employ single-view retrieval mechanisms for retrieval, and sometimes take advantage of ColBERT and other late-interaction approaches for re-ranking. Few research works have explored this domain in the scope of dimension size reduction, routing for learning ANNS, but these efforts are still nascent (Dhulipala et al., 2024; Santhanam et al., 2022; Ji et al., 2024; Kumar et al., 2023).

This paper introduces a novel routing-based approach to alleviate this limitation drawing inspiration from this vast literature of methods. We propose FastLane, a class of models upon which late-interaction methods such as ColBERT can now "pick" the most informative view for a given query, rather than exhaustively combining information from all views through the ["CLS"] token or other aggregation tricks. This can dramatically reduce computation without sacrificing performance as shown in Section 4.1. Our method learns to dynamically route each query to its most informative representation, enabling seamless integration with ANNS libraries and paving the way for highly scalable, accurate, and efficient multi-view retrieval systems.

**Motivation.** As discussed earlier, the limitation of late-interaction models such as ColBERT involves the non-parallelizable operation bottlenecked by the sum-max aggregation logic which limits its integrability with off-the-shelf ANNS. ANNS were not designed to for this paradigm of storing a pool of document embeddings for the same document index, and computing their score through sum-max operations. FastLane however picks a single query view and alleviates this need for computing the score between all query-document token representations, and allows standard ANNS such as Faiss, ScaNN to retrieve the most relevent document embeddings on the fly to compare against. Thus, we aim to bridge this gap by enabling the integration of late-interaction models with ANNS, thus improving both retrieval efficiency and result quality. In this work, we explore the extreme setting of "picking" a single optimal or informative representation, thus making the approach parallelizable. The implications of such an approach opens up possibilities into serving for longer query context lengths, and better models by default. We propose FastLane, a methodology to implement late-interaction models in practice through dynamic view routing. This routing mechanism is trained jointly with the encoder, and leads to an end-to-end differentiable pipeline similar to EHI (Kumar et al., 2023). To the best of our knowledge, FastLane is the first end-to-end learning method for late-interaction dense retrieval search. FastLane is seamlessly compatible with Approximate Nearest Neighbor Search (ANNS).

**Overview and Evaluation.** FastLane leverages the same backbone architecture as ColBERT (Khattab & Zaharia, 2020) and introduces a self-attention layer (Vaswani, 2017) over the final outputs of the query head. This layer generates logits for "picking" each view of the query, which are then transformed into probabilities with the help of the Gumbel-Softmax reparameterization trick (Jang et al., 2016). Furthermore, during training, we take advantage of the straight-through estimator to learn which distinct "views" or tokens to route to. Thus, FastLane helps eliminate the summation logic in the late-interaction operation while gradients being backprobable through all the tokens during training. We conduct a comprehensive empirical evaluation of our method against state-of-the-art techniques on standard benchmarks such as MS MARCO (Bajaj et al., 2016) and TREC-DL 19 (Craswell et al., 2020) (see Appendix A). Our experiments on the MS MARCO benchmark demonstrate that FastLane provides a speedup of up to approximately 30x in terms of computational complexity and 8x in terms of latency while maintaining comparable performance in terms of Mean Reciprocal Rank (MRR@10) and normalized discounted cumulative gain (NDCG@10). Moreover, FastLane significantly outperforms previous state-of-the-art single-view dense retrieval approaches by upto 8.14%. We observe a similar trend when working with the TREC DL-19 benchmark.

**Contributions.** This work presents FastLane, a novel approach for accelerating late-interaction dense retrieval with dynamic view routing. Our key contributions include:

- **Efficient Late-Interaction Retrieval with ANNS Compatibility.** FastLane eliminates the summation operation in late-interaction scoring through a learnable routing function, enabling parallelization and seamless integration with Approximate Nearest Neighbor Search (ANNS). This reduces computational complexity by up to **30x** while maintaining retrieval effectiveness. (see Section 5)
- **Strong Empirical Performance.** We conduct extensive evaluations on the MS MARCO and TREC-DL benchmarks, demonstrating that FastLane achieves competitive performance against state-of-the-art (SOTA) methods, including ColBERT, SGPT, ANCE, and DyNNIBAL, while significantly improving efficiency. (see Section 4.1, Appendix C)
- **Model-Agnostic Design.** While implemented with a ColBERT backbone, FastLane is adaptable to various encoder architectures, similarity metrics, and hard negative mining strategies, making it broadly applicable across dense retrieval frameworks. (see Section 3)

## 2 Related Work

The field of information retrieval has witnessed remarkable advancements, progressing from single-vector representations to more nuanced late-interaction methods that capture the complexities of queries and documents. These developments can be briefly categorized as (i) Single-Vector Retrieval Models, (ii) Late-Interaction Models, (iii) Static Models, and (iv) Dynamic Learnable Models for Routing.

**Single-Vector Retrieval Models.** Recent advances in contrastive learning (Gutmann & Hyvärinen, 2010) helped power strong dual encoder-based dense retrievers (Ni et al., 2021; Izacard et al., 2021; Nayak, 2019). These dual-tower models consist of query and document encoders, often shared, which are trained with contrastive learning using limited positively relevant query and document pairs (Menon et al., 2022; Xiong et al., 2020). These models condensed semantic information into the [CLS] token, enabling compatibility with Approximate Nearest Neighbor Search (ANNS) (Johnson et al., 2019). ANNS facilitated efficient retrieval by clustering the corpus and limiting the search space to smaller, relevant subsets of the original data. However, the simplicity of single-vector embeddings failed to accurately capture the rich, multi-dimensional semantics of complex queries and documents (Neelakantan et al., 2022; Muennighoff, 2022). This limitation motivated the development of retrieval systems with token-level granularity, laying the groundwork for such models.

**Late-Interaction Models.** Late-interaction models, as introduced through ColBERT (Khattab & Zaharia, 2020), represent a exploration of a new paradigm of models in retrieval. By introducing the use of token-level embeddings and a *sum-max* aggregation mechanism, these models utilized the ability to handle fine-grained semantic relationships between queries and documents. This approach significantly improved retrieval precision, making late-interaction models well-suited for nuanced information retrieval tasks. Building on this, ColBERTv2 (Santhanam et al., 2021), SPLADE (Formal et al., 2022) enhanced efficiency by refining the late-interaction mechanism and introducing lightweight document representations. However, we note

that despite efficiency-oriented improvements, multi-vector dense retrieval methods such as COLBERTv2, PLAID still incur nontrivial latency and memory costs due to multi-vector indexing, while sparse methods such as SPLADE involve different trade-offs related to inverted index size and expansion and both remain non-integrable with ANNS.

**Static Model Paradigm.** Due to the high memory, and energy requirements of late-interaction methods, their wide-spread adoption have remained limited. Workarounds in the form of two-stage pipelines helped reduce the load to the late-interaction models by re-ranking to fewer documents. However, ANNS were not designed to adapt to this change in paradigm. Multiple works have explore this dual-stage setup from a pre-trained baseline ColBERT setup. PLAID (Santhanam et al., 2022) introduced a multi-stage pipeline, progressively narrowing down candidate documents to minimize computational overhead. Along tangential lines, MUVERA (Dhulipala et al., 2024) transformed multi-vector similarity search into single-vector similarity search through Fixed Dimensional Encodings (FDEs), improving compatibility with ANNS solvers while maintaining retrieval accuracy. However, the cost of retrieving fewer views came with an increased dimension size, which increases the ANNS complexity linearly, while the number of views only increased the latency in a logarithmic fashion (as discussed briefly in Section 3). These approaches demonstrated the feasibility of scaling token-level interactions to larger datasets and complex retrieval scenarios, but remained disjoint steps.

**Dynamic Model Paradigm.** As retrieval tasks grow in complexity, trainable/learnable scoring models were introduced in Lite ColBERT (Ji et al., 2024). Lite ColBERT (Ji et al., 2024) combined the sum-max aggregation logic to a single attention scorer, followed by an MLP for representing the final similarity score. The Lite model helps reduce the number of tokens overall, making late-interaction models more practical during optimization. However, the aggregation would still require us to build a new paradigm of ANNS models for the task. We believe End-to-End Hierarchical Index (EHI) (Kumar et al., 2023) is the closest work in this literature which works with routing as an effective strategy to scale dense retrieval systems. By structuring the search space hierarchically, EHI significantly reduces computational costs, making them highly compatible with large-scale ANNS frameworks.

On a parallel effort, we introduce FastLane, a retrieval paradigm of models that bridges the efficiency of dual-tower models with the accuracy of late-interaction methods. By employing a differentiable routing mechanism optimized alongside the embedding model, FastLane dynamically identifies the most informative representations for retrieval. This selective routing reduces computational complexity by 30x (evaluated on queries with up to 30 tokens) while maintaining competitive accuracy. Moreover, FastLane seamlessly integrates with off-the-shelf ANNS frameworks and dynamic frameworks such as EHI (Kumar et al., 2023), providing a versatile solution for both existing and emerging retrieval systems. To the best of our knowledge, EHI remains the closest work to FastLane, where EHI works on routing to train a retrieval model in an end-to-end fashion. In this work, we extend the paradigm of EHI to bridge late-interaction models and current architectures.

## 3 Background

**Problem Definition and Notation.** Imagine we have a collection of $N$ queries, $\mathcal{Q} = \{q_1, \ldots, q_N\}$, and $M$ documents, $\mathcal{D} = \{d_1, \ldots, d_M\}$. Each query-document pair $(q_i, d_k)$ has a label $y_{ik}$, where $y_{ik} = 1$ means the document is relevant to the query, and $y_{ik} = -1$ means it is not. The goal is to build a system (called a *retriever*) that can take any query and find the most relevant documents from the collection.

Dense embedding-based retrieval (Johnson et al., 2019; Nayak, 2019; Dahiya et al., 2021; Khattab & Zaharia, 2020; Menon et al., 2022; Neelakantan et al., 2022; Muennighoff, 2022) is the state-of-the-art (SOTA) approach for semantic search and typically involves embeddings the documents and queries alike using a deep networks like BERT (Devlin et al., 2018).

**Single-View Approaches.** Single-view approaches (Menon et al., 2022; Neelakantan et al., 2022; Muennighoff, 2022) often involve a forward pass through a transformer to obtain a single embedding representation for both the query and document using the CLS token. The objective is to align the query CLS representation with the document CLS representation. Note that $W_s$ is typically a projection matrix used to map the BERT output into a shared embedding space for computing similarity score. In our experiments, we train a series

of BERT-based ColBERT models, and regular dual encoder models on the ("small") triplets training set, employing 6-layer BERT models from (Turc et al., 2019). Assuming a fixed maximum token length of m for queries and n for documents, the similarity score between a given query and document can be calculated as follows:

$$
\begin{aligned}
\hat{q} &= \texttt{BERT}([CLS; q_{1:m}])_1 \cdot W_s, \\
\hat{d} &= \texttt{BERT}([CLS; d_{1:n}])_1 \cdot W_s, \\
s &= \hat{q} \cdot \hat{d}.
\end{aligned}
\tag{1}
$$

However, single-view embeddings can struggle with queries that span multiple nuanced interpretations. For instance, consider the query "what is the cost of the apple?" This query might relate to the stock price of the company , or the fruit. Despite their obvious semantic difference, the propagation of which information users wish to ask for re-brands the information learned by the models. Thus, a single-view representation may fail to capture these diverse connections, grouping the query closer to a concept while missing the broader context. The latency to serve for these models via an off-the-shelf ANNS such as SCANN (Guo et al., 2020), FAISS (Johnson et al., 2019), or other hierarchical methods is $\mathcal{O}(n \log d)$, where $n$ is number of dimensions, and $d$ is the number of documents in our corpus. Such bottlenecks would also apply for MUVERA.

**Multi-View Approaches.** Late-interaction based (Khattab & Zaharia, 2020) methods which uses the representative power of multiple tokens addresses the single-view approach by accounting for the various semantics. The intuition as we understand it is that given a query, and the documents, those representations extracted from each token (through attention-based mechanisms to account for longer contexts) of the query are deemed relevant for representation learning, and thus retrieval. Assuming a fixed maximum token length of m for queries and n for documents, the similarity score between a given query and document can be calculated as follows:

$$
\begin{aligned}
\hat{q}_i &= \texttt{BERT}([CLS; q_{1:m}])_i \cdot W_s, & \mathcal{Q} &= \langle \hat{q}_0, \hat{q}_1, \ldots, \hat{q}_m \rangle, \\
\hat{d}_i &= \texttt{BERT}([CLS; d_{1:n}])_i \cdot W_s, & \mathcal{D} &= \langle \hat{d}_0, \hat{d}_1, \ldots, \hat{d}_n \rangle, \\
s &= \sum_{i=0}^{m} \max_{d_j \in \mathcal{D}} \langle \hat{q}_i, \hat{d}_j \rangle.
\end{aligned}
\tag{2}
$$

Although the approach aggregates via the Sum-Max late interaction, the increased capacity of the search space, the same also makes the approach slow as it no longer parallelizable due to the global synchronization barrier, which limits the adoption of these models and adaptable by off-the-shelf indexers which are primarily catered to single-view models. Furthermore, even if we deploy an ANNS for each view of the query, one would have a serving cost of $\mathcal{O}(v_{\text{query}} \cdot n \cdot \log(d \cdot v_{doc}))$, where $v_{\text{query}}$, and $v_{\text{doc}}$ are the number of tokens in the query, and the document respectively. $n$ is the number of documents, and $d$ is the dimension size. As noted from the asymptotic notations above for the ColBERT model, the computation complexity increases linearly in terms of the context-length of the query, and logarithmically in terms of the context-length of the document.

### 3.1 Overview of FastLane

Before delving further, we provide the intuition behind our approach to ground our findings in a simpler conceptual framework. People naturally possess a unique authorship style, shaped by their experiences and various influencing factors (Cheng et al., 2011). These styles, while distinct, often adhere to the foundational principles and structures of the language being used, as observed in prior work (Kumar et al., 2020). Languages inherently exhibit redundancies, such as stopwords, which act as connectors but do not typically drive the core intention of a query or text. We hypothesize that these redundancies create natural clusters in the token-space, where tokens sharing similar semantic or structural roles are likely to group together. This clustering serves as the basis for our approach.

We analyzed the token views generated by ColBERT model pre-trained on the LoTTE benchmark (see code [1]). We hypothesized that many tokens share semantically similar meanings, given that even with as many as 30 tokens, a typical query likely represents only 4-6 distinct semantic interpretations or views. Our intuition inspired by prior works could provide useful in the domain of retrieval.

As illustrated in Figure 2, our analysis confirms this intuition, revealing that many token representations exhibit high correlations, resulting in an interesting scenario where few unique clusters remain. By applying agglomerative clustering with a stringent threshold of 0.95, we observe very few distinct clusters (approximately 4-6 distinct representations). Consequently, the sum-max operation essentially reduces to a weighted average of a small subset of views. By selectively picking the most common query view through our scoring mechanism, we can bypass the summation process and seamlessly integrate multi-view retrieval with ANNS.

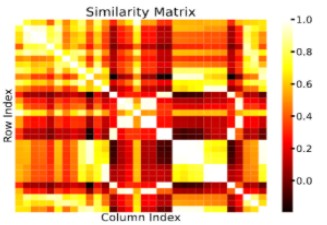 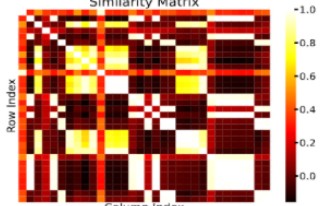 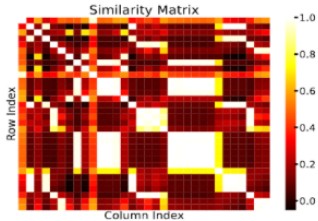

(a) How to tell the difference between a girl and boy bearded dragon?

(b) Is my corn snake male or female?

(c) Will betta fish eat snails?

Figure 2: The figure illustrates the token-level similarity matrix for a query, highlighting how the queries can be semantically clustered into a small number (4-6) of distinct token representations. Note that the cosine-similarity between embeddings is computed over subword tokens including special tokens such as CLS, SEP and padding, which explains the apparent increased number of representations in the heatmap plots.

## 3.2 Our Proposed Approach

Given this confirmation in our hypothesis, we aim to design a routing based multi-view search where only a given "view" (note that we will now starting to use "view", and "token representation" interchangably) of query is required to approximate the overall score. Due to redundancy of many of these tokens, we hypothesize that there might be an intrinsic signal to help learn this efficiently.

Our proposed system introduces a novel mechanism to optimize the retrieval process. Specifically, given the $M$ token embeddings of a query derived from a transformer model, we first pass these embeddings through a Self-Attention Layer followed by a Dense Layer. This pipeline generates a single probability value using a softmax activation function, that allows us to estimate the informativeness of each representation. Intuitively, if a specific token view achieves a higher probability, it signifies that the view captures more critical information about the query-document relationship. To enable efficient learning in this probabilistic framework, we incorporate the Gumbel-Softmax reparameterization trick (Jang et al., 2016), that ensures that the selection process remains differentiable. Additionally, we employ a straight-through estimator to maintain an end-to-end training pipeline, inspired by the approach in EHI (Kumar et al., 2023). This mechanism as proposed in EHI allows us to dynamically select a single representation from the $v_{\text{doc}}$ candidate views of a document during retrieval as well (see Figure 3 for an overview). While our current implementation focuses on selecting one view per query, future extensions could explore selecting multiple diverse views, potentially capturing multilingual or multi-modal representations—a direction we highlight as an exciting avenue for further research. Our approach introduces minimal overhead to the retrieval process while maintaining computational efficiency. The overall complexity is bounded by $\mathcal{O}(n \cdot \log(d \cdot v_{\text{doc}}))$, where $n$ is the number of dimensions, $d$, is the number of documents in the corpus, and $v_{\text{doc}}$ is the number of views per document which we would need to index. Furthermore, we could also significantly reduce the number of documents to index as depicted in Figure 2. This would significantly reduce the memory footprint of the model if we could

---

[1]https://github.com/stanford-futuredata/ColBERT

store only few pertinent (capturing most information or diverse views). This reduced memory bottleneck is crucial for serving to consumers at the scale of billions or trillions.

Furthermore, FastLane has a performance cost of $\mathcal{O}(n \cdot \log(d \cdot v_{doc}))$, a significant speedup of $v_{\text{query}}$ over ColBERT (as described in Section 3), and maintains performance (as shown in Section 4.1). This opens up interesting avenues in the domains of multi-lingual retrieval, long-context retrieval, multi-modal retrieval, and the current interests in retrieval augmented generation (Lewis et al., 2020).

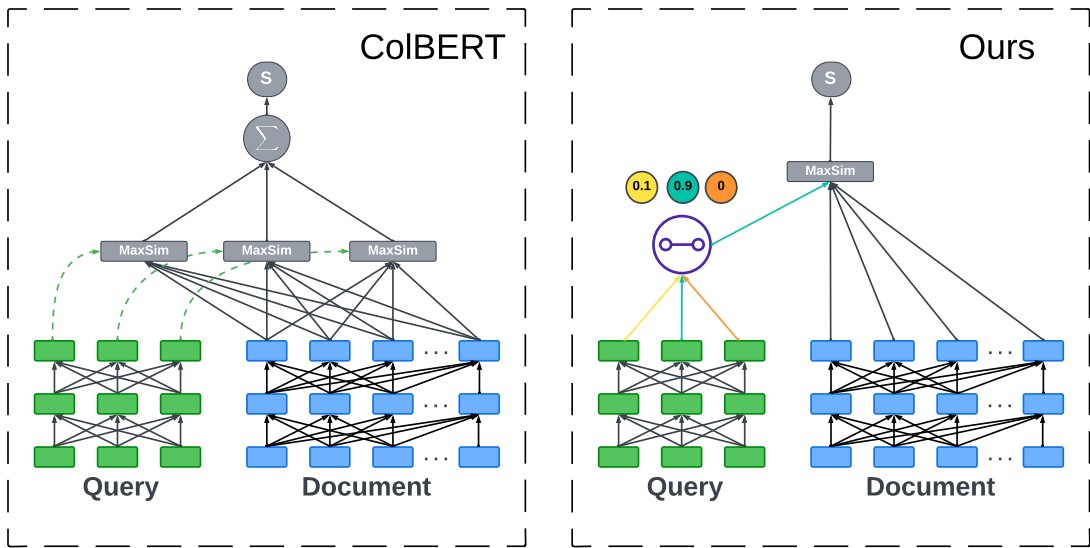

Figure 3: Comparison between ColBERT, and FastLane paradigm of models.

The FastLane model incorporates a multi-view retrieval strategy that leverages a self-attention mechanism to determine the importance of different views, followed by a probabilistic selection process using the Gumbel softmax reparameterization. The model is trained end-to-end, with a masking approach that regularizes view selection.

**Self-Attention Mechanism**   The FastLane model begins by taking multiple embedding representations as returned by the backbone ColBERT model (Khattab & Zaharia, 2020). For each view, the model applies a self-attention mechanism (Vaswani, 2017) to compute the relative importance of that view in the context of retrieval. Let the input token embeddings be represented as $v_1, v_2, \ldots, v_n$ where each $v_1$ is the representation of the first token, and so on. The self-attention mechanism computes attention scores for each view based on their pairwise interactions, as follows:

$$A_{ij} = \frac{Q(v_i) \cdot K(v_j)}{\sqrt{d_k}} \tag{3}$$

where $Q(v_i)$ and $K(v_j)$ are query and key transformations (restricted to single layer MLP in our case) respectively, and $d_k$ is the dimensionality of the key vectors. The attention scores are then passed through a softmax layer to normalize them across all the tokens $\alpha_{ij} = \text{softmax}(A_{ij})$. This produces a distribution over the different tokens, where each token is learned to approximate the "relevance to the retrieval task" as we progress through training.

**Gumbel Softmax Reparameterization**   To make the view selection process differentiable, the FastLane model employs the Gumbel-Softmax reparameterization (Jang et al., 2016) trick. The Gumbel-Softmax is a continuous approximation of discrete categorical sampling that allows backpropagation through the sampling process. To briefly summarize the process: Initially, a Gumbel noise $g_i$ is added to the logits of

the view probabilities $\alpha_i$, which perturbs the logits to encourage exploration $g_i = -\log(-\log(u_i))$, where $u_i \sim \text{Uniform}(0, 1)$ are random samples from the uniform distribution. The perturbed logits are then passed through the softmax function:

$$\hat{\alpha}_i = \frac{\exp((\log(\alpha_i) + g_i)/\tau)}{\sum_j \exp((\log(\alpha_j) + g_j)/\tau)}, \tag{4}$$

where $\tau$ is the temperature parameter controlling the smoothness of the distribution. As $\tau \to 0$, the distribution approaches a one-hot categorical distribution, and mimics picking. Furthermore, we note that other activation functions are also possible in this setting, including but not limited to Gumbel-Tanh, or using diverse sampling schemes such as Determinantal Point Processes (DPP; (Kulesza et al., 2012)) but such efforts remain beyond the scope of the current work, and are promising research avenues.

**Straight-Through Estimator**  To further facilitate seamless training and inference with the same architecture (similar to off-policy training, and on-policy evaluation), FastLane uses the straight-through estimator (STE) (Bengio et al., 2013). The STE allows gradients to propagate through the Gumbel softmax by treating the hard one-hot vectors as "continuous" during the forward pass and "discrete" during the backward pass. This enables the model to maintain end-to-end differentiability while performing discrete view selection. During training, the view with the highest score is selected though the straight-through estimator which picks the arg-max value over the softmax output. In the backward pass, gradients are passed through the continuous softmax outputs instead of the discrete arg-max, ensuring that the gradient flow remains intact.

**Masking and Regularization**  The FastLane model incorporates a masking mechanism to help select the view. The mask based on the view probabilities, ensuring that views with less information are masked out during training. The mask is a binary vector $m_i \in 0, 1$, where a value of 0 indicates that the corresponding view is masked out. To avoid trivial solutions, the mask is applied with a small constant $\epsilon$ that ensures smoother convergence $\alpha_i = \alpha_i \cdot (1 - m_i) + \epsilon.$, and prevents the model from completely ignoring certain views. The added $\epsilon = 0.05$ in our models acts as a regularizer, encouraging the model to explore during training.

**End-to-End learning**  The model is trained end-to-end. The self-attention mechanism learns to weight the importance of different views. The Gumbel-Softmax allows for differentiable selection of a single view. The STE enables gradient flow through the discrete selection process. The masking mechanism, combined with the small constant $\epsilon$, regularizes the learning, preventing the model from overly relying on a single view and promoting exploration of different views. The loss function compares the score of the selected view against the ground truth relevance, driving the model to learn informative view selection for efficient retrieval.

## 4  Experiments

In this section, we depict how our proposed approach (FastLane) compares to SOTA single-view dense retrieval approaches (Menon et al., 2022; Neelakantan et al., 2022; Muennighoff, 2022), and other late-interaction methods such as ColBERT (Khattab & Zaharia, 2020). We train a series of BERT-based ColBERT models, and regular dual encoder models on the ("small") triplets training set, employing 6-layer BERT models from (Turc et al., 2019). We train all models from scratch for 1.5 million steps until convergence. We also use the KL-distillation and obtain slightly better performance from our ColBERT models (Hofstätter et al., 2020). We report comparisons against these baselines on the MS MARCO and the TREC DL-19 benchmark as depicted below. Other numbers were taken as is from prior reports of results as collated in (Kumar et al., 2023; Menon et al., 2022). Our experimental setup is similar to the setup proposed in (Menon et al., 2022).

### 4.1  Results

Figure 4 depicts that FastLane is competitive to the ColBERT benchmark while being significantly faster. Appendix C depicts a more detailed comparison with additional baselines such as SGPT, cpt-text, ANCE, DyNNIBAL, DSI amongst others. Furthermore, parallel research endeavours around negative mining,

distillation and others remain an open and interesting avenue for such models. As showcased in Table 3, FastLane achieves competitive performance in comparison to SOTA and is servable at the latency as single-view dual-encoder models as showcased in the table. This is an improvement of approximate **8.14%** in MRR@10, **6.4%** in nDCG@10 on MS MARCO benchmarks, and **1.6%** in nDCG@10 on the TREC DL-19 benchmark as further depicted in Table 1. This paradigm of learning a routing function for query representations while maintaining the advantages and increased capacity of these multi-view approaches, helps place FastLane, as a step towards wider adoption of these models, and interesting research directions towards longer context lengths of queries, multi-lingual retrieval models, multi-modal retrieval models, and many more.

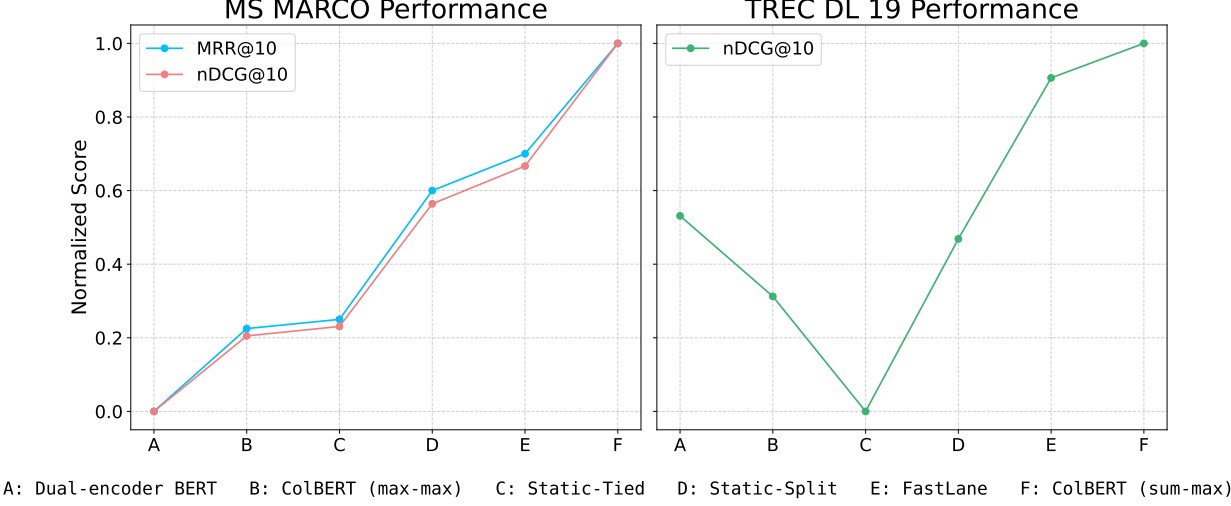

A: Dual-encoder BERT    B: ColBERT (max-max)    C: Static-Tied    D: Static-Split    E: FastLane    F: ColBERT (sum-max)

Figure 4: Relative performance of different models on MS MARCO and TREC DL-19 datasets, showing improvement over the Dual-encoder BERT baseline and normalized between 0 and 1.

**Routing Decision.** To understand the effect various routers, we design experiments comparing FastLane against another retrieval algorithms with varying routing intuitions. In this work, the goal has been to choose a single view for the query. Naively, this could be done when one resorts to only using the "CLS" token output for query (average of all views without focussing on the relative importance of each of them). This would essentially keep the cost similar to FastLane. However, as showcased in Table 1, `Static-Tied` (with tied towers where query, and document towers share weights), this is shown to be worse off in comparison to FastLane. Furthermore, we create another ablation where we do not tie the towers and do note a significant improvement as depicted in Table 1, titled `Static-Split` (with different towers where query, and document towers do not share weights). This could be viewed analogous as using only the ["CLS"] token from the query model, while using all the token representations in the corpus. As showcased in Table 1, we note that FastLane outperforms these dual-encoder (single-view) models as well as alternative static routing schemes through learning a dynamic routing mechanism.

Table 1: Performance metrics of MRR@10 and nDCG@10 evaluated on the MS MARCO dev set and TREC DL-19. The best result for each metric is shown in **bold**, and the second-best is marked with †.

| Model | **MRR**@10 (MS MARCO) | **nDCG**@10 (MS MARCO) | **nDCG**@10 (TREC DL-19) |
|---|---|---|---|
| Dual-encoder BERT | 0.344 | 0.404 | 0.742 |
| ColBERT (max-max) | 0.353 | 0.412 | 0.735 |
| Static-Tied | 0.354 | 0.413 | 0.725 |
| Static-Split | 0.368 | 0.426 | 0.740 |
| FastLane | $0.372^{\dagger}$ | $0.430^{\dagger}$ | $0.754^{\dagger}$ |
| ColBERT (sum-max) | **0.384** | **0.443** | **0.757** |

| Method | Index Size | Avg Latency (s) | p90 Latency (s) | Throughput (QPS) | CPU RAM (Peak) | GPU Mem (Peak) |
|---|---|---|---|---|---|---|
| ColBERT (sum-max) | 28.61 GiB | 1.221 | 1.222 | 81.9 | 32.93 GiB | 37.78 GiB |
| FastLane (1-view) | 28.61 GiB | **0.064** | **0.064** | **1557.5** | 33.00 GiB | **28.78 GiB** |

Table 2: Comparison of retrieval latency, throughput, and resource usage between ColBERT (sum–max) and FastLane under identical indexing and hardware settings.

**Latency gains.** Conventional late-interaction models (including ColBERT (Khattab & Zaharia, 2020)) traditionally use the sum-max aggregation, and are not readily integrable with Approximate Nearest Neighbor Search (ANNS). Thus, these models are not readily deployed in practice. Instead, practitioners often use single-view dual-encoder models in practice, and sometimes rely on ColBERT models for re-ranking. In this work, we proposed a methodology to overcome this two-stage process and combining these steps into a single pipeline. FastLane achieves this by "picking" a view of query to use, and eliminating the summation part in our computation. We consider a corpus of $100k$ documents, where each document is represented by 200 token-level embeddings of dimension 768, and queries consist of 30 token embeddings. All experiments are performed using FP16 precision on a single A6000 GPU, and both methods operate over the same document index, ensuring identical index size (28.61 GiB) and comparable CPU memory footprint. Retrieval latency is measured end-to-end for scoring queries against the full corpus, and we additionally report p90 latency to capture tail behavior. We further include throughput (queries per second), peak CPU resident memory (RSS), and peak GPU memory allocation to reflect deployment-relevant constraints. Importantly, the only difference between the two methods lies in the query–document interaction: while ColBERT employs a sum-max aggregation over all query tokens, FastLane selects a single informative query view and eliminates the summation, thereby reducing the computational complexity of retrieval. The results, summarized in Table 2, shows that FastLane achieves a 19x reduction in average and tail latency, together with a corresponding 19x increase in throughput, while maintaining the same index size and nearly identical CPU memory usage. In addition, FastLane reduces peak GPU memory allocation by approximately 9 GiB, highlighting its practical advantage for large-scale and resource-constrained retrieval settings.

## 5 Limitations and Future Work

In this section, we justify the various design choices, and lay the groundwork for few interesting future research endeavors around the paradigm of FastLane models.

**Memory Bottlenecks** One limitation, that remains unsolved with the current approach is significantly larger the memory footprint of late-interaction approaches such as ColBERT (Khattab & Zaharia, 2020). Note that a cost of indexing in single-view dual-tower models is $d$ documents of the corpus. However, in late-interaction models such as ColBERT, one would need to index $\mathcal{O}(d \cdot v_{\text{doc}})$, where $d$ is the number of documents, and $v_{\text{doc}}$ is the number of document tokens or views. This number is usually large, around 200 tokens in our experiments, but could be much larger as well. Some works including (Santhanam et al., 2021) works upon quantization and reduction of dimensions of the embeddings. We note this as a potential direction to help integrate FastLane with. Another option is to cluster these view embeddings per document and use a Determinantal Point Processes (DPP) or clustering followed by picking diverse views to help reduce memory footprint of our multi-view models as shown in Figure 2.

**Future Works.** Through our study as depicted in Table 3, we notice an interesting avenue of future research endeavor where FastLane could facilitate the use arbitrarily longer context-length in the query side. This is further corroborated from our results as depicted in Table 1. This improved performance of FastLane over single-view dual encoder approaches shows promise not only for enabling more powerful algorithms in our general systems, but also enables the use of arbitrarily longer context length in the query side (adhering to memory and fairness constraints). The multi-view embedding space, even if sparsely accessed, provides a richer representation capacity. This allows for finer-grained semantic distinctions, potentially capturing nuances that the single-view encoder struggles to represent (improving expressiveness, while reducing computational cost). The view selection mechanism acts as a dynamic "switch", adapting to different queries and selecting the most suitable subspace within the larger multi-view embedding space. This provides flexibility that a fixed single-view encoder lacks. Such an approach opens up avenues for research of integrating with CLIP (Radford

et al., 2021) to account for multiple languages and facilitate significantly longer query context lengths for retrieval. This would have direct implications on Multi-lingual retrieval, Multi-modal retrieval, and Retrieval Augmented Generation (Lewis et al., 2020).

## 6    Conclusion

In this work, we introduce FastLane, a learnable routing paradigm that enhances late-interaction models like ColBERT by balancing accuracy and efficiency. By dynamically scoring and selecting the most informative token-level representations, FastLane significantly reduces the computational complexity of multi-view retrieval systems. This enables FastLane to bridge the gap of late-interaction models and off-the-shelf ANNS, making it a scalable and practical solution for large-scale retrieval tasks. FastLane has an intrinsic ability to handle longer context lengths efficiently for retrieval. By improving the scalability, adaptability, and computational efficiency of these late-interaction models, FastLane represents a shift towards a new-paradigm of dynamic routed models.

**Ethical Broader Impact Statement**

This paper introduces FastLane, a method to improve the scalability and efficiency of late-interaction retrieval models, specifically by enabling dynamic routing. Our contributions advance the state-of-the-art in retrieval by significantly reducing computational overhead while maintaining competitive performance metrics. The societal impact of this work is primarily centered on improving the accessibility and practicality of information retrieval systems, especially in large-scale applications such as search engines, recommendation systems, and question-answering platforms. By reducing latency and memory costs, this method can enable the deployment of more efficient systems, making advanced retrieval technology accessible to broader audiences, including resource-constrained settings. From an ethical standpoint, we note that retrieval models are not free from potential biases in their training data. While FastLane reduces computational costs, it inherits any biases present in the underlying data and models, which could impact its retrieval decisions. We encourage further research on bias mitigation in retrieval systems to enhance fairness and inclusivity. While there are potential societal consequences of this work, we do not identify specific risks or ethical concerns that require immediate mitigation beyond the aforementioned considerations.

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

## A  Datasets

In this section, we briefly discuss the open-source datasets used in this work adapted from a similar setup as Menon et al. (2022).

**MS Marco**  The MS Marco benchmark (Bajaj et al., 2016) has been included since it is widely recognized as the gold standard for evaluating and benchmarking large-scale information retrieval systems (Thakur et al., 2021; Ni et al., 2021). It is a collection of real-world search queries and corresponding documents carefully curated from the Microsoft Bing search engine. What sets MSMarco apart from other datasets is its scale and diversity, consisting of approximately 9 million documents in its corpus and 532,761 query-passage pairs for fine-tuning the majority of the retrievers. Due to the increased complexity in scale and missing labels, the benchmark is widely known to be challenging. The dataset has been extensively explored and used for fine-tuning dense retrievers in recent works (Thakur et al., 2021; Nogueira & Cho, 2019; Gao et al., 2020; Qu et al., 2020). MSMarco has gained significant attention and popularity in the research community due to its realistic and challenging nature. Its large-scale and diverse dataset reflects the complexities and nuances of real-world search scenarios, making it an excellent testbed for evaluating the performance of information retrieval algorithms.

**TREC DL-19**  The TREC Deep Learning 2019 (TREC DL-19) benchmark (Craswell et al., 2020) is a well-established evaluation suite designed to assess the performance of retrieval systems in the context of deep learning-based search. It builds upon the MS MARCO passage ranking task by using a subset of its queries, enriched with high-quality relevance judgments from the TREC expert assessors. The benchmark comprises 43 queries derived from MS MARCO, with relevance annotations that are significantly more granular and reliable compared to automatic weak labels. TREC-DL 19 is particularly valued for its human-annotated labels and its emphasis on ranking quality, making it suitable for robust evaluation of fine-grained retrieval capabilities. Due to its curated annotations and widespread adoption in the IR community, the dataset is frequently used as a downstream evaluation benchmark to validate the real-world effectiveness of retrievers (Thakur et al., 2021). It serves as a critical diagnostic tool for testing generalization and transfer capabilities, especially in zero-shot and low-resource settings.

## B  Model

In line with prior work (Menon et al., 2022), we implement and train a diverse set of BERT-based retrieval architectures to evaluate the impact of static and dynamic routing strategies in multi-view document representations.

**Model Architecture.**  For all retrievers, we use a 6-layer small-BERT encoder initialized from the pre-trained models of (Turc et al., 2019), with hidden size 768 and 12 attention heads. The document encoder produces contextualized token embeddings, while the query encoder either pools or scores these token-level embeddings depending on the retrieval architecture. All models are trained from scratch for 1.5 million steps until convergence, without using any large-scale in-domain pretraining or contrastive bootstrapping. For dual encoder models, query and document encoders are either shared (tied) or instantiated independently (untied), depending on the specific model variant. ColBERT-based models compute similarity via late interaction using a max or sum pooling over token-level inner products (Santhanam et al., 2021).

**Model Variants.**  We predominantly compare the following retrieval architectures, but also compare against other State-of-the-Art models for the sake of completeness as depicted in Appendix C. We list few of these model variants below:

- **Dual-encoder BERT**: A baseline model with tied query and document encoders and a single ["CLS"] token used for computing dot-product similarity. This serves as the canonical dual encoder setup which is typically deployed in production.

- **ColBERT (max-max)**: A ColBERT-style model using max pooling on both query and document token representations.

- **Static-Tied**: A static routing baseline where the query and document encoders are tied. The first view (or "CLS" token) is always selected as the query representation, making it a single-view static retriever with minimal capacity.

- **Static-Split**: Similar to Static-Tied, but the query and document encoders are untied. This configuration increases model capacity slightly while retaining a static routing policy.

- **FastLane (ours)**: Our full model with learned dynamic routing over multiple query views. The model selects the most relevant view for each query through a learned router, trained end-to-end alongside the retriever. Query and document encoders are untied, and token-level matching is performed using a sum-aggregate scoring.

- **ColBERT (sum-max)**: A stronger late-interaction variant where token-level similarities are aggregated using sum over queries and max over documents, yielding the best overall performance.

**Training Details.** All models are trained with AdamW (Loshchilov & Hutter, 2017) using a linear warmup and cosine decay scheduler. We tune learning rates in the range $\{5 \times 10^{-6}, 1 \times 10^{-5}, 3 \times 10^{-5}, 5 \times 10^{-5}\}$ separately for each model variant to ensure fair comparisons. Weight decay is set to 0.01 across our experiments. Following Hofstätter et al. (2020), we use KL-divergence based distillation from a strong BM25 teacher, which yields modest but consistent gains across all ColBERT-based variants as also reported by Menon et al. (2022).

## C  Additional Results

Table 3 depicts the main results of our experiments, comparing Fastlane to several SOTA baseline models. As shown in the table, Fastlane achieves competitive performance across both the MS MARCO and TREC DL-19 datasets. Specifically, Fastlane demonstrates notable improvements in MRR@10 and nDCG@10 compared to the Dual-encoder BERT and ColBERT (max-max) models. While Fastlane slightly underperforms ColBERT (sum-max) on both datasets, the difference is relatively small. We believe this slight performance trade-off is justified by the substantial gains in retrieval speed, as detailed in Section 4.1.

### C.1  Failure Cases and Limitations of Single-View Routing

FastLane is explicitly designed as an efficiency-oriented approximation of late-interaction retrieval. As such, it inherits a fundamental limitation: when a query requires multiple orthogonal semantic facets to be satisfied simultaneously, collapsing the query-document interaction to a single routed view can discard information that full sum–max aggregation would otherwise preserve. In our analysis, we observe two broad classes of queries for which FastLane is more likely to underperform full late-interaction models such as ColBERT.

**Multi-intent and conjunctive queries.** These queries contain multiple constraints that must be jointly satisfied, often corresponding to distinct query tokens (e.g., an entity combined with an attribute, condition, or temporal qualifier). In such cases, no single token embedding may adequately capture the complete intent of the query. Sum-max aggregation allows different query tokens to independently contribute evidence, yielding a more faithful relevance signal than a single routed representation.

**Polysemous or ambiguous queries.** Queries with inherently ambiguous terms may activate multiple semantic interpretations, each associated with different subsets of query tokens. Single-view routing may commit early to one interpretation, whereas late-interaction scoring can retain multiple semantic hypotheses through token-level interactions. These failure modes are not unique to FastLane; rather, they are a direct consequence of replacing a many-to-many interaction function with a single-vector proxy. Accordingly, FastLane should be understood not as a universal replacement for late interaction, but as a controllable point on the accuracy-latency trade-off curve. Finally, these observations naturally motivate several extensions, which we leave for future work: (i) *multi-view routing*, where a small set of token-level views is selected

instead of a single one; (ii) *conditional fallback mechanisms*, where queries deemed uncertain revert to full sum-max scoring; and (iii) *hybrid retrieval pipelines*, where FastLane is employed as a first-stage retriever followed by a more expressive late-interaction or cross-encoder re-ranking stage.

Table 3: Performance metrics of $MRR$@10 and $nDCG$@10 evaluated on the MS MARCO dev dataset and TREC DL-19 dataset. The best value for each metric is indicated in **bold**, and the second-best is marked with †.

| Model | MSMARCO re-rank | | TREC DL19 re-rank | #. parameters. |
|---|---|---|---|---|
| | MRR | nDCG | nDCG | |
| DPR (Thakur et al., 2021) | - | 0.177 | - | 110M (BERT-base) |
| BM25 (official) (Khattab & Zaharia, 2020) | 0.167 | 0.228 | 0.501 | - |
| BM25 (Anserini) (Khattab & Zaharia, 2020) | 0.187 | - | - | - |
| **cpt-text** S (Neelakantan et al., 2022) | 0.199 | - | - | 300M (cpt-text S) |
| **cpt-text** M (Neelakantan et al., 2022) | 0.206 | - | - | 1.2B (cpt-text M) |
| **cpt-text** L (Neelakantan et al., 2022) | 0.215 | - | - | 6B (cpt-text L) |
| **cpt-text** XL (Neelakantan et al., 2022) | 0.227 | - | - | 175B (cpt-text XL) |
| DSI (Atomic Docid + Doc2Query + Base Model) (Tay et al., 2022) | 0.260 | 0.3228 | - | 250M (T5-base) |
| DSI (Naive String Docid + Doc2Query + XL Model) (Tay et al., 2022) | 0.210 | - | - | 3B (T5-XL) |
| DSI (Naive String Docid + Doc2Query + XXL Model) (Tay et al., 2022) | 0.165 | - | - | 11B (T5-XXL) |
| DSI (Semantic String Docid + Doc2Query + XL Model) (Tay et al., 2022) | 0.203 | 0.2786 | - | 3B (T5-XL) |
| CCSA (Lassance et al., 2021) | 0.289 | - | 0.583 | 110M (BERT-base) |
| HNSW (Malkov & Yashunin, 2018) | 0.289 | - | - | - |
| RoBERTa-base + In-batch Negatives (Monath et al., 2023) | 0.242 | - | - | 123M (RoBERTa-base) |
| RoBERTa-base + Uniform Negatives (Monath et al., 2023) | 0.305 | - | - | 123M (RoBERTa-base) |
| RoBERTa-base + DyNNIBAL (Monath et al., 2023) | 0.334 | - | - | 123M (RoBERTa-base) |
| RoBERTa-base + Stochastic Negatives (Monath et al., 2023) | 0.331 | - | - | 123M (RoBERTa-base) |
| RoBERTa-base +Negative Cache (Monath et al., 2023) | 0.331 | - | - | 123M (RoBERTa-base) |
| RoBERTa-base + Exhaustive Negatives (Monath et al., 2023) | 0.345 | - | - | 123M (RoBERTa-base) |
| SGPT-CE-2.7B (Muennighoff, 2022) | - | 0.278 | - | 2.7B (GPT-Neo) |
| SGPT-CE-6.1B (Muennighoff, 2022) | - | 0.290 | - | 6.1B (GPT-J-6B) |
| SGPT-BE-5.8B (Muennighoff, 2022) | - | 0.399 | - | 5.8B (GPT-J) |
| doc2query (Khattab & Zaharia, 2020) | 0.215 | - | - | - |
| DeepCT (Thakur et al., 2021) | 0.243 | 0.296 | - | 110M (BERT-base) |
| docTTTTquery (Khattab & Zaharia, 2020) | 0.2077 | - | - | - |
| SPARTA (Thakur et al., 2021) | - | 0.351 | - | 110M (BERT-base) |
| docT5query (Thakur et al., 2021) | - | 0.338 | - | - |
| DeepImpact (Mallia et al., 2021) | 0.326 | 0.385 | 0.695 | 110M (BERT-base) |
| ANCE | 0.330 | 0.388 | - | - |
| RepCONC (Zhan et al., 2022) | 0.340 | - | - | 123M (RoBERTa-base) |
| Dual-encoder BERT (6-layer) (Menon et al., 2022) | 0.344 | 0.404 | 0.742 | 67.5M (small-bert) |
| Cross-attention BERT (12-layer) (Menon et al., 2022) | - | - | 0.749 | 110M (BERT-base) |
| DistilBERT + MSE (Menon et al., 2022) | - | - | 0.693 | 66M (distilBERT) |
| TAS-B (Thakur et al., 2021) | - | 0.408 | - | 110M (BERT-base) |
| GenQ (Thakur et al., 2021) | - | 0.408 | - | - |
| ColBERT (re-rank) (Khattab & Zaharia, 2020) | 0.348 | - | - | 110M(BERT-base) |
| ColBERT (end-to-end) (Khattab & Zaharia, 2020) | 0.360 | 0.40 | - | 110M (BERT-base) |
| BM25 + CE (Thakur et al., 2021) | - | 0.413 | - | - |
| SPLADE (simple training) (Formal et al., 2022) | 0.342 | - | 0.699 | 110M (BERT-base) |
| DistilBERT + Margin MSE (Menon et al., 2022) | - | - | 0.718 | 66M (distilBERT) |
| DistilBERT + RankDistil-B (Menon et al., 2022) | - | - | 0.708 | 66M (distilBERT) |
| DistilBERT + Softmax CE (Menon et al., 2022) | - | - | 0.726 | 66M (distilBERT) |
| DistilBERT + $M^3$SE (Menon et al., 2022) | - | - | 0.714 | 66M (distilBERT) |
| SPLADE + DistilMSE (Formal et al., 2022) | 0.358 | - | 0.729 | 66M (distilBERT) |
| SPLADE + SelfDistil (Formal et al., 2022) | 0.368 | - | 0.723 | 66M (distilBERT) |
| SPLADE + EnsembleDistil (Formal et al., 2022) | 0.369 | - | 0.721 | 66M (distilBERT) |
| SPLADE + CoCondenser-SelfDistil (Formal et al., 2022) | 0.375 | - | 0.73 | 66M (distilBERT) |
| SPLADE + CoCondenser-EnsembleDistil (Formal et al., 2022) | 0.380 | - | 0.732 | 66M (distilBERT) |
| FastLane (**Ours**) | 0.372 | 0.430† | 0.754† | 67.5M (small-bert) |
| ColBERT (sum-max) | 0.384† | **0.443** | **0.757** | 67.5M (small-bert) |
| ColBERT-v2 (Santhanam et al., 2021) | **0.397** | - | - | 110M (BERT-base) |

