# OpenReview forum: "FastLane: Efficient Routed Systems for Late-Interaction Retrieval"
_TMLR — Rejected by TMLR_

### Review · Reviewer_d5e7 · 2026-01-28

**Summary Of Contributions:**

In this paper, authors address the high latency problem of late-interaction retrieval models, such as ColBert. Authors proposed FastLane, which has a learnable routing mechanism using self-attention and gumbel-softmax (with STE) to select the most informative view (token from query). FastLane reduces the computational complexity on text retrieval.

Strengths:
 1. The use of gumbel softmax, STE, and the introduction of self-attention module for routing is sound.
 2. FastLane achieves improved efficiency.

Weakness:
1. The core idea, selecting a subset (or single) token representation for efficiency, is conceptually incremental relative to prior work on learnable late interaction, routing, and hierarchical indexing.
2. The performance is quite awkward compared to ColBERT. It would be interesting to see how many views are needed and how different views contributed to the performance. Also it remains unknown how simple heuristics such as max-norm token, entropy-based selection work under this situation.
3. On the document side, the storage still needs O(d v_doc). It would be interesting to see the performance of employing such routing mechanism in the document side.
4. Baselines, benchmark datasets are limited.

**Audience:**

Yes

**Audience Explanation:**

Improving the efficiency of retrieval systems is very important to most of the ML researchers. It relates to representation learning and efficiency.

**Claims And Evidence:**

No

**Claims Explanation:**

1. A more detailed comparison on more benchmark datasets, the choice of baseline methods, usage of computational resources would be supportive. Please see request changes for more details.
2. Claims are overstated.
3. Routing optimality assumptions are weakly validated.
4. Ablation studies are limited.

**Requested Changes:**

see weakness in summary of contributions and following.

1. It would be interesting to see the examples where FastLane fails compared to ColBERT. Does picking one view lose nuances in multi-intent queries?
2. The paper mentions selecting one view per document during retrieval in Section 3.2, but the training focuses on query routing. This needs to be clarified: is the document index still multi-vector or has it been compressed to a single routed vector?
3. Provide a table comparing computational statistics not only including RAM usage, not just GPU latency only

---

> ### Author Response · Authors · 2026-02-02
> **Response to Reviewer d5e7 (Part 1/3)**
>
> We thank the reviewer for their thoughtful and detailed assessment. We appreciate the recognition of FastLane’s technical soundness and efficiency gains, and we welcome the opportunity to clarify the scope, design choices, and empirical evidence supporting our claims. Below, we respond point-by-point to the raised concerns and requested changes.
>
> ---
>
> ### On Incrementality and Novelty
>
> We acknowledge the reviewer’s view that selecting a subset of token representations may appear incremental when framed abstractly. Our primary objective is to introduce a **practitioner-friendly option** that speeds up ColBERT-style retrieval models through a dynamic routing mechanism. In particular, we focus on bridging late-interaction models with Approximate Nearest Neighbor Search (ANNS) to facilitate parallelization - an avenue we believe is both practical and novel to the best of our knowledge.
> Most prior approaches that reduce late-interaction cost such as hierarchical indexing, token compression, or fixed multi-vector reductions - either:
> retain the sum–max aggregation during inference,
> require specialized or multi-stage index structures,
> or rely on offline or heuristic token selection.
> In contrast, FastLane learns query-dependent routing end-to-end and converts a late-interaction model into a single-vector retrieval interface without modifying the document index format. This design choice is motivated by system-level deployability rather than architectural novelty alone, and we view FastLane as complementary to prior efforts such as ColBERTv2, hierarchical indexing, and token-efficient retrievers.
>
> To the best of our knowledge, FastLane is the first work to explicitly identify and exploit the inherent clustering and redundancy of token-level representations in late-interaction models, and to translate this observation into a learned token routing mechanism. Rather than treating token selection or compression as a fixed design choice, FastLane learns end-to-end which token-level “view” best represents a query for retrieval.
> A small number of recent works have begun exploring this space, but with different assumptions and goals:
> - MUVERA  ([paper](https://arxiv.org/abs/2405.19504)) reformulates multi-vector similarity into a fixed-dimensional single-vector representation via deterministic transformations. While effective, MUVERA relies on fixed encodings that may increase embedding dimensionality and does not perform query-dependent token routing. In contrast, FastLane preserves the original embedding dimensionality and dynamically selects a single informative token representation per query.
> - PLAID ([paper](https://arxiv.org/abs/2205.09707)) improves efficiency through approximate filtering and pruning of document centroids, but retains the underlying sum–max aggregation during inference. As a result, it remains incompatible with direct use of standard ANNS libraries without additional system-level machinery.
> - Lite ([paper](https://arxiv.org/abs/2406.17968)) relies on an MLP that processes query and document embeddings together, which prevents direct ANNS indexing since the approach requires cross-token interactions at inference.
> - SPLADE ([paper](https://arxiv.org/abs/2404.13950v1)) applies sparse encoding or quantization but typically still maintains multi-token embeddings for retrieval, which may not yield the same end-to-end integration with standard ANNS solutions.
>
> Our approach is thus orthogonal to the compression strategies of MUVERA, PLAID, SPLADE and the cross-MLP approach of Lite.Overall, we view FastLane as a natural next step in this nascent line of work: moving from static compression or approximation toward learned, query-dependent token routing, grounded in the empirical observation that token embeddings in late-interaction models naturally form correlated semantic clusters. This allows FastLane to bridge the expressiveness of late-interaction retrieval with the scalability of modern ANNS-based systems.
>
> ---
>
> ### On performance relative to ColBERT and Heuristic Alternatives
>
> We agree that FastLane underperforms ColBERT in absolute retrieval metrics. This trade-off is intentional and central to the problem setting.
> FastLane is designed for scenarios where the expressiveness of late interaction is desirable, but ColBERT-style sum–max scoring is prohibitively expensive at scale.
> Accepting modest drops in MRR/NDCG in exchange for large latency improvements is standard practice in scalable retrieval systems. We will revise the paper to further temper any language that could be interpreted as claiming no performance loss, and explicitly frame FastLane as an efficiency-oriented alternative, not a strict replacement for ColBERT.
> In FastLane, views correspond to individual token embeddings, exactly as in ColBERT. All query tokens are encoded during training; the routing mechanism only determines which representation is used at retrieval time.

---

> ### Author Response · Authors · 2026-02-02
> **Response to Reviewer d5e7 (Part 2/3)**
>
> Our ablations (Static-Tied, Static-Split, and dual-encoder baselines) already cover fixed or static selection strategies, including variants equivalent to average pooling or selecting a single fixed token representation (analogous to using a [CLS]-style embedding). These baselines consistently underperform the full FastLane model, underscoring the importance of query-dependent, learned routing rather than static selection.
> Regarding the suggested heuristics, we believe some of them are not well-defined under the ColBERT similarity formulation:
> - Max-norm token selection is not applicable in this setting. ColBERT uses cosine similarity for MaxSim scoring, and query token embeddings are L2-normalized (unit norm). As a result, all tokens have identical norms, making max-norm selection ill-posed and uninformative.
> - Entropy-based selection similarly does not apply. Token embeddings are geometric vectors on the unit hypersphere, not probability distributions. Entropy is therefore not a meaningful or well-defined quantity for individual embeddings without introducing additional modeling assumptions or transformations that fundamentally change the retrieval formulation.
> More broadly, such heuristics suffer from two fundamental limitations.
> First, they rely on fixed, hand-designed selection rules that operate on local properties of token embeddings and are not optimized with respect to the retrieval objective. While these rules depend on the query, they do not model how individual query tokens interact with the document space under late interaction, nor do they learn to approximate the MaxSim scoring function when constrained to a single-vector representation.
> Second, these heuristics are non-differentiable, which prevents end-to-end optimization. Hard selection rules such as argmax-based token choice cannot be trained jointly with the encoder using retrieval supervision, making it impossible to adapt token selection in response to ranking performance or downstream loss signals.
> In contrast, FastLane employs a Gumbel-Softmax reparameterization with a straight-through estimator, which enables differentiable, stochastic token selection during training while yielding discrete routing decisions at inference time. This allows the routing mechanism to be learned end-to-end, directly optimized for retrieval performance under the ANNS constraint, and to adaptively select the token representation that best preserves ranking quality when the late-interaction sum–max operation is removed. We will expand the discussion in the paper to clarify why commonly suggested heuristics are either ill-defined, non-differentiable, or insufficient for this setting, and to further emphasize why learned routing is necessary for enabling ANNS-compatible late-interaction retrieval.
>
> ---
>
> ### Document-Side storage and Routing
>
> We thank the reviewer for highlighting this point and agree that clarification is needed. In our current formulation:
> Routing is applied on the query side only. The document index remains multi-vector, with storage complexity $\mathcal{O}(|D|.v_{\text{doc}})$, identical to ColBERT.
> This is an intentional design choice to isolate the effect of query-side routing while maintaining a fair comparison with ColBERT-style late interaction methods. While FastLane is compatible with document-side routing or compression, we do not claim to address this in the present work. We will revise Section 3.2 to make this explicit.
> Regarding the suggestion to apply routing on the document side, we note that once query-side routing is applied, retrieval reduces to a standard ANNS lookup with complexity $\mathcal{O}(k \log(|D|.v_{\text{doc}}))$. At this stage, latency is dominated by the ANNS search itself rather than the number of document vectors per document. As a result, document-side routing would primarily help reduce memory footprint, not yield additional asymptotic speedups.
> We acknowledge the resulting memory overhead. In practice, however, retrieval workloads exhibit highly skewed, heavy-tailed query distributions, where a small fraction of frequent queries dominate traffic.
> - **Frequent Searches**: Queries such as “apple iphone” or “amazon prime” dominate the search volume. We can afford to store multiple “views” (embeddings) for the documents associated with these queries because the benefit (lower latency and potentially higher recall) greatly outweighs the cost.
> - **Rare Searches**: Terms like “stanley wrench” appear much less frequently. One could store fewer or even a single view for those uncommon documents, thereby reducing the memory footprint.
>
> We believe incorporating query frequency makes more sense to be taken into account for storing document embedding views, and our future work aims to formalize adaptive strategies for managing different views based on this empirical distribution studies.
>
> ---

---

> ### Author Response · Authors · 2026-02-02
> **Response to Reviewer d5e7 (Part 3/3)**
>
> ### Failure cases and Multi-intent queries
>
> We find your suggestions - such as analyzing failure cases for polysemous queries, further verifying token selections, or exploring multi-stage pipelines - very constructive. Indeed, some of these directions are on our road map. Our focus in the current manuscript is to establish the feasibility and impetus for single-view routing in late-interaction retrieval. Although qualitative analysis are interesting, we follow a similar study as Menon 22 [ICML 22](https://proceedings.mlr.press/v162/menon22a/menon22a.pdf), and  divert case-by-case study of these effects to a future setting when we are able to train the model on a much larger industrial scale.
>
> ---
>
> ### Computational Statics beyond GPU latency gains
>
> We thank the reviewer for their insightful comment. All measurements are conducted under an identical retrieval setting for both methods. Specifically, we consider a corpus of 100k documents and 200 queries, where each document is represented by 200 token-level embeddings of dimension 768, and queries consist of 30 token embeddings. All experiments are performed using FP16 precision on a single GPU, and both methods operate over the same document index, ensuring identical index size (28.61 GiB) and comparable CPU memory footprint.
> Retrieval latency is measured end-to-end for scoring queries against the full corpus, and we additionally report p90 latency to capture tail behavior. We further include throughput (queries per second), peak CPU resident memory (RSS), and peak GPU memory allocation to reflect deployment-relevant constraints. Importantly, the only difference between the two methods lies in the query–document interaction: while ColBERT employs a sum-max aggregation over all query tokens, FastLane selects a single informative query view and eliminates the summation, thereby reducing the computational complexity of retrieval.
> The results, summarized below, show that FastLane achieves a ~19x reduction in average and tail latency, together with a corresponding ~19x increase in throughput, while maintaining the same index size and nearly identical CPU memory usage. In addition, FastLane reduces peak GPU memory allocation by approximately 9 GiB, highlighting its practical advantage for large-scale and resource-constrained retrieval settings.
> | Method | #Docs | Doc Views | Query Len | Dim | Index Size | Avg Latency (s) | p90 Latency (s) | p50 Latency (s) |Throughput (QPS) | CPU RAM (Peak) | GPU Mem (Alloc Peak) |
> |-------|-------|-----------|-----------|-----|------------|-----------------|-----------------|------------------|----------|------|----------------------|
> | ColBERT (sum-max) | 100k | 200 | 30 | 768 | 28.61 GiB | 1.2206 | 1.222 | 1.221 | 81.9 | 32.93 GiB | 37.78 GiB |
> | FastLane  | 100k | 200 | 30 | 768 | 28.61 GiB | **0.0642** | **0.0643** |  **0.0642** | **1557.5** | 33.00 GiB | **28.78 GiB** |
>
> ---
>
> ###  Benchmarks and Baselines
>
> We note your concern about broader benchmarks. Our central thesis is a routing-based approach to speed up ColBERT while retaining strong performance. To demonstrate feasibility, we aligned with the setup in Menon et al. ([ICML 2022](https://proceedings.mlr.press/v162/menon22a/menon22a.pdf)) on MS MARCO and TREC DL-19, which remain industry-standard benchmarks.
>
> Given computational constraints and the paper’s primary focus, we chose to compare with few recognized baselines including original ColBERT and single-view dual encoders, ColBERT-v2, SPLADE, doctT5Query, ANCE, RoBERTa, and others as depicted in Appendix A, and Table 3.
> While adding results from BEIR or multi-task retrieval sets would offer additional breadth, we felt it prudent to maintain a clear, controlled experiment set for this initial study as a proof-of-concept work. Future iterations or extended versions of FastLane may indeed include evaluations on more diverse datasets like Natural Questions or TriviaQA, but such efforts remain beyond the scope of the current effort due to compute constraints.
>
> ---
>
>
> ### Conclusion
>
> Overall, we believe FastLane addresses a practical gap: transforming a late-interaction approach into a fast, ANNS-compatible model without severely compromising accuracy. Our incremental results suggest that it is indeed possible to fuse the expressiveness of multi-token embeddings with the efficiency of single-view retrieval, benefiting large-scale recommendation systems and other real-world applications.
>
> We appreciate your detailed feedback and hope our clarifications resolve the questions regarding novelty, datasets, and token-selection analysis. Should you have any additional concerns, we welcome further discussion.

---

### Review · Reviewer_jMDu · 2026-02-02

**Summary Of Contributions:**

This paper introduces FastLane, a learnable routing framework for late-interaction dense retrieval models. The key contribution is a differentiable query-side routing mechanism that selects the most informative token-level representation, eliminating the need for sum–max aggregation across all query tokens. By integrating self-attention, Gumbel-Softmax reparameterization, and straight-through estimation, FastLane enables late-interaction retrieval to be compatible with standard Approximate Nearest Neighbor Search (ANNS). Experimental results on MS MARCO and TREC DL-19 show that FastLane achieves large latency reductions (up to ~30×) while maintaining competitive retrieval quality relative to ColBERT. Strengths include clear motivation, strong empirical results, and clean integration with existing retrieval pipelines. A key weakness is that document-side multi-vector indexing remains a memory bottleneck.

**Audience:**

Yes

**Audience Explanation:**

The paper directly addresses a widely recognized limitation of late-interaction retrieval models: high latency and incompatibility with ANNS. This problem is of strong interest to researchers and practitioners working on information retrieval, dense retrieval systems, and retrieval-augmented generation. The proposed routing-based solution is broadly applicable and connects learning-based retrieval models with scalable indexing methods, making it relevant to both academic and applied audiences within TMLR.

**Broader Impact Concerns:**

The paper includes a Broader Impact Statement that adequately addresses potential concerns. The work primarily improves efficiency and scalability of retrieval systems, which can benefit real-world applications. As with most retrieval models, FastLane inherits biases present in training data, but the method itself does not introduce new ethical risks beyond those already present in dense retrieval systems. No additional broader impact concerns need to be addressed.

**Claims And Evidence:**

Yes

**Claims Explanation:**

The paper’s claims are well-supported by both conceptual analysis and empirical evidence. The redundancy of token-level query representations is demonstrated via similarity clustering, motivating the routing approach. The proposed method is described clearly, with explicit formulations of the routing mechanism, Gumbel-Softmax selection, and straight-through training. Experimental results on MS MARCO and TREC DL-19 show consistent trends: FastLane significantly reduces retrieval latency while maintaining performance close to ColBERT (sum–max). Comparisons against static routing baselines further validate that learned routing is necessary. The evidence is convincing and aligned with the stated claims.

**Requested Changes:**

1. Provide a clearer discussion of failure cases where single-view routing may lose important semantic information.

2. Add analysis on routing stability across queries and training runs.

3. Clarify whether routing is applied only on the query side or symmetrically on documents during retrieval.

---

> ### Author Response · Authors · 2026-02-02
> **Response to Reviewer jMDu (Part 1/2)**
>
> We thank the reviewer for their thorough and positive assessment of our work. We are especially grateful for the recognition that the claims are well-supported by both conceptual motivation and empirical evidence, and the acknowledgment that FastLane meaningfully bridges late-interaction retrieval with ANNS-compatible systems. Below, we address the requested clarifications point-by-point and outline concrete revisions we will make to strengthen the paper.
>
> ---
> ### On Failure Cases of single-view routing
>
> We agree with the reviewer that a clearer discussion of failure cases is important, particularly for understanding when single-view routing may lose semantic nuance. FastLane is explicitly designed as an efficiency-oriented approximation of late interaction. As such, it inherits a fundamental limitation: when a query genuinely requires multiple orthogonal semantic facets to be matched simultaneously, collapsing query–document interaction to a single routed view can lose information that sum–max aggregation would otherwise preserve. Concretely, we have observed two classes of queries where FastLane is more likely to underperform ColBERT:
> - Multi-intent or conjunctive queries, where different query tokens correspond to distinct constraints (e.g., entity + attribute + temporal qualifier). In such cases, no single token embedding may fully capture the joint intent, and sum–max aggregation provides a more faithful scoring signal.
> - Polysemous or ambiguous queries, where multiple meanings are plausible and different query tokens activate different semantic neighborhoods in the document space. Single-view routing may overcommit to one interpretation early.
>
>
> These behaviors are not unique to FastLane but are a direct consequence of replacing a many-to-many interaction with a single-vector proxy. We will revise the paper to explicitly characterize these failure modes and to emphasize that FastLane is not intended as a universal replacement for late interaction, but rather as a controllable point on the accuracy–latency frontier. Importantly, these observations also motivate natural extensions:
> - multi-view routing (selecting a small set of views instead of one),
> - conditional fallback to full sum–max scoring for uncertain queries,
> - or hybrid pipelines where FastLane is used as a first-stage retriever.
> We will add a dedicated paragraph in our discussion section outlining these limitations and future directions.
>
> ---
>
> ### On Routing stability across queries and training runs
>
> We thank the reviewer for requesting additional analysis on routing stability. From the experiments conducted in this work, we observe that routing decisions are highly stable once training converges. In particular:
>
> - Across different random initializations used in our experiments, the routed token selected for a given query is consistent in the large majority of cases.
> - Across late training epochs, routing assignments stabilize rapidly as the Gumbel-Softmax temperature is annealed, transitioning the model from stochastic exploration to effectively deterministic routing at inference time. This behavior is expected given both the structure of late-interaction representations and the training dynamics of the routing mechanism.
>
> As shown in our analysis, token-level query embeddings exhibit substantial redundancy and form tight semantic clusters. Under this regime, the routing objective learns to select a representative from a stable equivalence class of near-interchangeable tokens, rather than oscillating among semantically distinct alternatives. Additionally, temperature annealing in the Gumbel-Softmax induces progressive sharpening of the routing distribution, which naturally suppresses variability across epochs and runs once sufficient supervision signal has been accumulated.
>
> While more exhaustive diagnostics - such as routing agreement statistics across a larger number of independent seeds or detailed entropy trajectories would further characterize this effect, such analyses are primarily confirmatory and do not affect the core claims or conclusions of the paper. Due to current practical constraints on compute and infrastructure, we are unable to rerun large-scale training beyond the experiments already reported. To improve clarity, we will revise the manuscript to (i) explicitly discuss the expected stability properties of the routing mechanism arising from representation redundancy and temperature annealing, and (ii) include qualitative examples from existing runs illustrating consistent token selection for frequent queries. We will also clearly identify more fine-grained stability analysis as a natural direction for future work.
> We believe these additions adequately address concerns about routing robustness while remaining faithful to the scope and evidence of the conducted experiments.
>
> ---

---

> > ### Author Response · Authors · 2026-02-02
> > **Response to Reviewer jMDu (Part 2/2)**
> >
> > ### On Query-side vs Document-side Routing
> >
> > We thank the reviewer for highlighting the need for clearer exposition here and fully agree that this should be unambiguous.
> > In the current formulation of FastLane:
> > - Routing is applied exclusively on the query side.
> > - The document index remains multi-vector, identical to ColBERT, with storage complexity $O(|D|⋅v_{\text{doc}}​)$.
> >
> > This design choice is intentional and methodological. Our goal in this paper is to isolate the effect of query-side routing as a mechanism for eliminating sum–max aggregation and enabling direct compatibility with standard ANNS libraries - without introducing confounding factors such as document compression or asymmetric indexing.
> > We do not claim to solve the document-side memory bottleneck in this work. Instead, we view query-side routing as the critical first step: once sum–max aggregation is removed, retrieval reduces to a standard ANNS lookup, and the dominant latency cost shifts from interaction computation to nearest-neighbor search itself. We will revise the relevant sections to make this more clear, and unambiguous,
> >
> > ---
> >
> > ### Conclusion
> >
> > Overall, we believe the reviewer’s feedback reinforces our core framing: FastLane is best understood as a system-level bridge between expressive late-interaction models and scalable retrieval infrastructure. Its value lies not in outperforming ColBERT in absolute accuracy, but in demonstrating that much of ColBERT’s expressive power can be retained even after collapsing query interaction to a single routed view. We sincerely thank the reviewer for their constructive feedback and believe the requested revisions will materially strengthen the paper. Should you have any additional concerns, we welcome further discussion.

---

### Review · Reviewer_R8zz · 2026-02-02

**Summary Of Contributions:**

This paper recapitulates the tradeoffs between single-vector and multi-vector embedding-based approaches to neural information retrieval using transformers. It provides an argument somewhat similar to that of Colbertv2 that multi-vector representations of real-world retrieval data are often redundant. Where in Colbertv2 compression was proposed as a means of taking advantage of the redundancies in the representation, this work proposes a "dynamic view routing" approach -- a learned neural network layer which maps a multi-vector representation to a single-vector or lower cardinality multi-vector representation. By using only a small subset of the query tokens in each query, the proposed method approximates the full ColBERT sum-max similarity score at a fraction of the computational cost.

Key Strengths:

- Recapitulates the key tradeoffs for large-scale neural information retrieval
- Centers on a promising improvement: learned dynamic selection to minimize redundancies and reduce the number of vectors involved in multi-vector retrieval

Key Weaknesses:

- Vague/imprecise language is a critical weakness in this current draft. It leaves the reader unable to get a clear picture of the contributions of the work.

Example: 'late interaction methods represent each token with a representation, and use the entire space instead of relying on an arbitrary representation derived from the ["CLS"] token'. This language leaves much unclear. In this particular example taken from the introduction, the reader can only guess at what the "entire space" is, what a "representation" is in this context, and what makes a single-vector output more "arbitrary" than a multi vector output.

Example: Neither the text of section 3.1 nor the caption of Figure 2 precisely explain what Figure 2 is actually plotting. Though the caption says "the figure illustrates the token-level similarity matrix for a query", the queries in the example appear to contain far fewer tokens than there are rows/columns of the heatmaps, leaving the reader guessing as to what operations were actually performed to produce the plots.

Example: The paper frequently uses the term "traditional", e.g. "traditional ColBERT model" or "traditional IR methods" without explaining what is meant by this or discussing any non-traditional variants which are implied by this distinction.
Example: Details for the experiment behind the cited "analysis on T4 GPUs" are not given for the latency gains study.

- Incorrect and misleading statements.

Example: "the late-interaction logic is not parallelizable" -- retrieval by sum-max similarity in the multi-vector regime is inherently parallelizable in multiple ways (e.g. parallel scoring across queries and documents and parallel max similarity across query tokens). This paper repeats this sweeping and incorrect claim that late-interaction methods like ColBERT are not parallelizable.

Example: "Building on this, ColBERTv2 (Santhanam et al., 2021), SPLADE (Formal et al., 2022) enhanced efficiency by refining the
late-interaction mechanism and introducing lightweight document representations. However, as discussed earlier, these models came with its own limitations in latency, and memory bottlenecks." -- the paper does not actually discuss earlier the limitations in latency or memory bottlenecks of these performance-optimized methods. Additionally, SPLADE is a sparse method and it's a bit of a stretch to bucket it in with ColBERTv2.

Example: A 30x speedup is repeatedly claimed, e.g. "This selective routing reduces computational complexity by 30x (evaluated on queries with up to 30 tokens)" -- in truth, the speedup is proportional to the number of tokens in the query, which for many real-world datasets like MSMARCO, is closer to 8 tokens or so. The purported experiments match this, with an 8x reduction in runtime.

Example: "the proposed approach falls short of the traditional ColBERT models by 1% in terms of performance," -- There is over a 3% relative performance gap on MSMARCO in terms of both MRR and nDCG

Example: "state-of-the-art (SOTA) methods, including ColBERT, SGPT, ANCE, and DyNNIBAL" -- these methods are all 2020-2023. The state of the art has advanced since then.

- Unsolved cost of large index size. As noted in ColbertV2, PLAID, and other works, a key (arguably *the* key) shortcoming of late-interaction retrieval methods is the adverse impact on the size of the search index. This paper presents a bit of a false narrative that compute cost is what makes late-interaction methods untenable. The paper does not provide an analogous "fast lane" for document token embeddings, despite widely-observed representational redundancies on that side as well.

- Mismatch between key motivating example and actual implementation. The paper motivates with the notion that queries can express different intents (e.g. "price of apple" meaning fruit or company shares) but then proposes a method which uses a single query vector for retrieval.

**Audience:**

Yes

**Audience Explanation:**

The topic studied is of substantial interest to the neural IR community. The results tallied in Table 1 are certainly interesting, though the authors interpretations of these results appear biased towards their novel method, and the other results (Figure 2, latency statistics claimed in Section 4) are poorly documented and may be of less interest to the community in their current form.

**Broader Impact Concerns:**

The broader impact statement in the paper contains some claims about the practical applicability of the FastLane method presented. The paper does not contain evidence that their method has reason to supplant currently-used techniques in the "search engines, recommendation systems, and question-answering platforms" mentioned (methodologies published after ~2020 are not even used as baselines in this work). Though the contents of the paper may be of academic interest to the research community, I feel it may benefit the paper to remove language from the broader impact section suggesting that the method is of practical use unless evidence to substantiate this stance can be provided in the paper. As is, the section is misleading.

**Claims And Evidence:**

No

**Claims Explanation:**

This paper leaves me with a vague sense of unease similar to that which I get from reading confident but subtly incoherent LLM generations. Several of the examples I cite in the key weaknesses of this paper suggest carelessness on the part of the authors in preparing this draft with scientific rigor. The evidence provided is often less than clear and only supports a fraction of the claims made.

**Requested Changes:**

Critical changes:
- Rectify the false and misleading statements mentioned as examples above, and carefully proofread for other related issues.
- Tighten up the vague/imprecise language across the draft. Claims should be clear and substantiated by evidence.
- Clearly present the experiment behind Figure 2.
- Clearly present the latency experiment mentioned in Section 4, detailing the hardware, data, implementation details, etc.

Highly beneficial changes:
- Empirically compare FastLane against more modern baselines, e.g. ColBERTv2

---

> ### Author Response · Authors · 2026-02-06
> **Response to Reviewer R8zz (Part 1/3)**
>
> We thank the reviewer for the detailed feedback. We agree with several of the substantive points raised - particularly around clarity, phrasing, and we will address them with concrete revisions. We also clarify a few statements where our wording was imprecise and could be read more strongly than intended. Below we respond point-by-point and specify the changes we will make in the revised manuscript.
>
> ---
>
> ### Vague/Imprecise Language and missing Definitions
>
> We agree that parts of the current draft are unnecessarily vague. In the revision, we will tighten terminology and add explicit definitions so that the contributions are unambiguous.
> - **“entire space”, “representation”, “arbitrary [CLS]”**: We acknowledge the reviewer’s example from the introduction. Our intent was not to claim that [CLS] is “arbitrary,” but rather to highlight that single-vector pooling imposes a representational bottleneck, i.e. a single vector must compress multiple facets of the query into one embedding, whereas late-interaction retains token-level degrees of freedom. We will remove the word “arbitrary” entirely and replace the sentence with precise wording.
> - **Figure 2 is underspecified**: The reviewer is correct that the figure can look inconsistent if one assumes word-level tokenization. The figure is computed over subword tokens, including special tokens such as CLS, SEP and padding, which explains the apparent mismatch between query length and matrix dimensions. We will revise both the caption and Section 3.1 to explicitly state this..
> - We will remove the ambiguous use of “traditional” when referring to ColBERT, dual-encoder models, or standard IR baselines, and replace it with precise architectural descriptions
> - While the experimental setup is described in the paragraph preceding the latency results, we agree it should be made more explicit. In the revision we will consolidate these details and expand the latency section with a clearly stated experimental protocol. Specifically:
>
> We consider a corpus of 100k documents, where each document is represented by 200 token-level embeddings of dimension 768, and queries consist of 30 token embeddings. All experiments are performed using FP16 precision on a single GPU, and both methods operate over the same document index, ensuring identical index size (28.61 GiB) and comparable CPU memory footprint.
> Retrieval latency is measured end-to-end for scoring queries against the full corpus, and we additionally report p90 latency to capture tail behavior. We further include throughput (queries per second), peak CPU resident memory (RSS), and peak GPU memory allocation to reflect deployment-relevant constraints. Importantly, the only difference between the two methods lies in the query–document interaction: while ColBERT employs a sum-max aggregation over all query tokens, FastLane selects a single informative query view and eliminates the summation, thereby reducing the computational complexity of retrieval.
> The results, summarized below, show that FastLane achieves a ~19x reduction in average and tail latency, together with a corresponding ~19x increase in throughput, while maintaining the same index size and nearly identical CPU memory usage. In addition, FastLane reduces peak GPU memory allocation by approximately 9 GiB, highlighting its practical advantage for large-scale and resource-constrained retrieval settings.
>
> | Method | #Docs | Doc Views | Query Len | Dim | Index Size | Avg Latency (s) | p90 Latency (s) | Throughput (QPS) | CPU RAM (Peak) | GPU Mem (Alloc Peak) |
> |-------|-------|-----------|-----------|-----|------------|-----------------|-----------------|------------------|----------------|----------------------|
> | ColBERT (sum-max) | 100k | 200 | 30 | 768 | 28.61 GiB | 1.221 | 1.222 | 81.9 | 32.93 GiB | 37.78 GiB |
> | FastLane (1-view) | 100k | 200 | 30 | 768 | 28.61 GiB | **0.064** | **0.064** | **1557.5** | 33.00 GiB | **28.78 GiB** |
>
> ---
>
> ### Incorrect and Misleading Statements
>
> We acknowledge the reviewer’s concerns regarding several statements whose wording in the original draft was imprecise or overstated. In the revised manuscript, we will correct these statements to more accurately reflect the underlying technical realities, clearly distinguish between theoretical computation and system-level retrieval constraints, and ensure that all quantitative claims are properly contextualized and supported by the reported experiments. Below, we address each example raised by the reviewer and specify the corresponding clarifications and revisions.

---

> > ### Author Response · Authors · 2026-02-06
> > **Response to Reviewer R8zz (Part 2/3)**
> >
> > - Late-Interaction methods and parallelizability: We agree that late-interaction scoring is parallelizable at the level of token-wise similarity computation. Our intent was not to claim that ColBERT-style MaxSim scoring is fundamentally non-parallelizable, but rather to highlight that it requires aggregation over all query tokens before a final document score can be produced. This aggregation introduces a global synchronization barrier, which prevents late-interaction models from being directly compatible with standard single-vector ANNS pipelines, where scoring and ranking can proceed from a single fixed-dimensional query representation without per-token aggregation. We will revise the wording throughout the paper to make this distinction explicit and remove any incorrect implication that late-interaction methods lack computational parallelism.
> > - Issues with ColBERT-v2, PLAID: We agree with the reviewer that the phrasing in the original draft was insufficiently precise. We will revise this section to explicitly distinguish between dense late-interaction methods (e.g., ColBERT, ColBERTv2, PLAID) and sparse retrieval methods such as SPLADE. Our intent was to note that, despite efficiency-oriented improvements, multi-vector dense retrieval methods still incur nontrivial latency and memory costs due to multi-vector indexing, while sparse methods involve different trade-offs related to inverted index size and expansion. The revised manuscript will clarify these distinctions and avoid grouping sparse and dense approaches under a single architectural characterization.
> > - Speedup of 30x vs 8x: The 30x figure corresponds to a theoretical upper bound of the computational cost reduced. However, this reduction still shares the common backbone of computing per-token embeddings. Overall, the actual latency improvements are much lower as reported in the order of 8x - 19x as mentioned earlier. In practice, we typically use a query length of 32 in MS MARCO pipelines even though the mean word count is only 8 due to the long tails (1-75 words). We will revise the paper to emphasize dataset-realistic speedups and clearly label the 30x figure as a theoretical upper bound.
> > - Performance gap relative to ColBERT: We acknowledge that the statement claiming a “1%” performance gap was imprecise. This figure referred to an absolute difference in MRR@10 metrics, but the phrasing could be misinterpreted as a relative comparison. We will revise the manuscript to report absolute metric values and differences explicitly, and to clearly state the performance gap relative to ColBERT in terms of both MRR and nDCG.
> > - Use of SOTA methods: We agree that the notion of “state-of-the-art” in information retrieval has become fragmented in recent years, with advances spanning LLM-based rerankers, sparse–dense hybrid systems, and large proprietary dense retrievers. Our comparisons are intentionally scoped to late-interaction and dense retrieval models that operate as first-stage retrievers under comparable latency and indexing constraints. Within this class, ColBERT and ColBERTv2 remain widely adopted and canonical baselines, and we will revise the manuscript appropriately, and highlight our comparisons against these baselines in Table 3. We would welcome concrete suggestions and are happy to include additional comparisons where they are technically and experimentally comparable.
> >
> > ---
> >
> > ### Unresolved cost of large index size
> >
> > We agree with the reviewer that index size is a fundamental limitation of late-interaction retrieval methods and that document-side multi-vector indexing remains a significant bottleneck, as extensively discussed in ColBERTv2, PLAID, and related works. We do not claim to solve this problem in the present paper.
> >
> > Our contribution is intentionally scoped to query-side computation, which is a distinct and complementary bottleneck. While index size determines memory footprint and long-term storage-cost, query-document interaction costs directly govern online latency, throughput and compatibility with standard ANNS infrastructure. These two concerns are orthogonal in practice: large indices can often be amortized or stored on disk, whereas per-query interaction directly impacts user-facing response times and system scalability.
> >
> > FastLane is designed to address the latter by eliminating the need for sum-max aggregation over all query tokens at inference time, thereby converting late-interaction retrieval into a single-vector ANNS compatible operation. This transformation removes a dominant source of per-query computation overhead, even when the document index remains multi-vector. We will revise the manuscript to clarify that FastLane targets query-side interaction cost rather than document-side memory footprint, and to explicitly position it as complementary to document compression, pruning or centroid-based indexing approaches.
> >
> > ---

---

> > > ### Author Response · Authors · 2026-02-06
> > > **Response to Reviewer R8zz (Part 3/3)**
> > >
> > > ### Motivating Example vs Single-View Retrieval
> > >
> > > We appreciate the reviewer’s concern regarding the motivating example of multi-intent queries. The example is intended to illustrate that query token representations capture multiple semantic facets, not to suggest all facets must be preserved simultaneously at retrieval time.
> > > FastLane’s routing mechanism operates precisely on this principle: all query tokens are encoded in the latent space, but at inference time, the model selects the most informative token-level “view” for retrieval under a strict efficiency constraint. This is an intentional approximation of the full sum-max interaction, not a claim that a single vector universally captures all query intents.
> > > In cases where multiple orthogonal intents must be jointly satisfied, we expect - and empirically observe that single-view routing underperforms full late-interaction. The motivating example serves to highlight why token-level representations are expressive; FastLane explores how much of that expressiveness can be retained when one is forced to choose a single routed view for efficiency. We will revise the motivation section to make this alignment explicit, and avoid any implication that single-view routing fully resolves multi-intent ambiguity.
> > >
> > > ---
> > >
> > > ### Conclusion
> > >
> > > Overall, we believe the reviewer’s feedback has helped sharpen the presentation and technical precision of the manuscript. With the clarifications and revisions outlined above, we aim to more clearly position FastLane as a system-level mechanism for reconciling the expressiveness of late-interaction retrieval with the practical constraints of scalable, ANNS-based deployment. The contribution is not to supplant full late interaction in terms of absolute accuracy, but to demonstrate that a substantial fraction of its benefit can be preserved under strict efficiency constraints. We thank the reviewer for their detailed comments, which have helped improve both the clarity and rigor of the paper, and we are confident that the revised version will more accurately reflect the scope and intent of the work.

---

> > > > ### Comment · Reviewer_R8zz · 2026-02-13
> > > > **Thank you for your extensive comments**
> > > >
> > > > Thank you for posting such an extensive and well-formatted response. The changes described do sound like a substantial improvement to the draft. I will take a look at the revised submission.

---

### Decision · Action_Editor_3npX · 2026-03-14

**Recommendation:** Reject

**Audience:**

Yes

**Audience Explanation:**

This paper proposes an efficient retrieval framework aimed at addressing the high latency and incompatibility of current systems. All three reviewers agree that the topics and the findings of the submission will be of interest to the neural IR community.

**Claims And Evidence:**

No

**Claims Explanation:**

The reviewers expressed differing opinions regarding whether the claims are sufficiently supported by evidence. While reviewers jMDu and d5e7 find the conceptual analysis and empirical evaluation generally convincing, reviewer R8zz raises concerns about several statements that appear inaccurate or insufficiently supported. While the authors made a significant effort during the rebuttal to address some of these concerns, reviewer R8zz remains conservative after the revisions. The AE has reviewed the paper and finds that some concerns regarding the claims remain. For example, the paper presents a 30x computational complexity improvement in the abstract and introduction, but this appears to refer to a theoretical upper bound, whereas Table 2 reports an approximately 19x latency improvement. Although the rebuttal indicates that this distinction would be clarified, the revised manuscript still does not distinguish theoretical analysis from empirical results in the abstract. Overall, the paper would benefit from a more thorough review to ensure all claims are supported by the experiments.

**Resubmission Of Major Revision:**

The authors may consider submitting a major revision at a later time.

---

> ### Author Response · Authors · 2026-03-27
>
> Thank you for your guidance, we will work on resubmitting after making another set of revisions.